# Bounding the Invertibility of Privacy-Preserving Instance Encoding Using Fisher Information

**Kiwan Maeng**[*]
Penn State University
kvm6242@psu.edu

**Chuan Guo**[*]
FAIR, Meta
chuanguo@meta.com

**Sanjay Kariyappa**
Georgia Institute of Technology
sanjaykariyappa@gatech.edu

**G. Edward Suh**
FAIR, Meta & Cornell University
edsuh@meta.com

## Abstract

Privacy-preserving instance encoding aims to encode raw data into feature vectors without revealing their privacy-sensitive information. When designed properly, these encodings can be used for downstream ML applications such as training and inference with limited privacy risk. However, the vast majority of existing schemes do not theoretically justify that their encoding is non-invertible, and their privacy-enhancing properties are only validated empirically against a limited set of attacks. In this paper, we propose a theoretically-principled measure for the invertibility of instance encoding based on Fisher information that is broadly applicable to a wide range of popular encoders. We show that dFIL can be used to bound the invertibility of encodings both theoretically and empirically, providing an intuitive interpretation of the privacy of instance encoding.

## 1 Introduction

Machine learning (ML) applications often require access to privacy-sensitive data. A model that predicts diseases with x-ray scans requires access to patient's x-ray images. Next-word prediction models require access to user's text input that can contain sensitive information [23]. To enable ML applications on privacy-sensitive data, *instance encoding* ([6]; Figure 1, left) aims to encode data in a way such that it is possible to run useful ML tasks—such as model training and inference—on the encoded data while the privacy of the raw data is preserved. The concept is widespread under many different names: private data sharing [16, 17, 48, 5, 34, 32, 37], learnable encryption [35, 73, 71, 70], split learning [63, 55], split inference [40, 11], and vertical federated learning (vFL; [74, 59, 45]) are all collaborative training/inference schemes that operate on (hopefully) privately-encoded user data.

Unfortunately, the vast majority of existing instance encoding schemes rely on heuristics rather than rigorous theoretical arguments to justify their privacy-enhancing properties. Most existing works [35, 73, 64, 65, 45] claimed that their schemes are non-invertible by simply testing them against certain input reconstruction attacks. However, these schemes may not be private under more carefully designed attacks; in fact, many encoding schemes that were initially thought to be private have been shown to be vulnerable over time [6, 7]. While there exist instance encoding schemes that are based on a rigorous theory, *e.g.*, metric differential privacy (metric-DP; [8, 16, 17, 44]), their theoretical analysis is only applicable to very simple encoders and cannot be applied to more powerful encoders (*e.g.*, DNN-based encoders) that are common in practice [73, 64, 65, 45, 5, 34, 32].

---

[*]Equal contribution.

37th Conference on Neural Information Processing Systems (NeurIPS 2023).

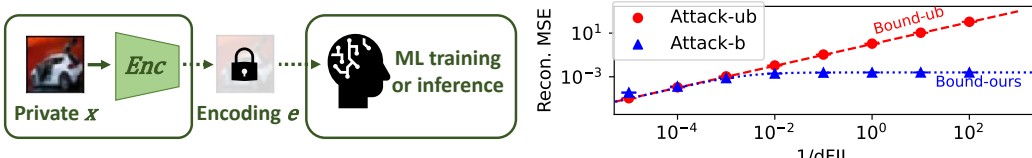

Figure 1: Overview of instance encoding (left), and our bound against different attackers (*Attack-ub*, *Attack-b*) for a synthetic dataset (right). The bound adapted from prior work [20] (*Bound-ub*) only works against a certain adversary (*Attack-ub*). Our newly proposed bound (*Bound-ours*) works for any adversary. See Section 3.3.1 for evaluation details.

To fill the gap, we propose a *theoretically-principled* and *easily applicable* framework for quantifying the privacy of instance encoding by bounding its invertibility. Our framework uses (diagonal) *Fisher information leakage* (dFIL; [18, 22, 20])—an information-theoretic measure of privacy with similar properties to differential privacy (DP; [14, 15]). dFIL can be computed for common privacy-enhancing mechanisms and can lower-bound the expected mean squared error (MSE) of an input reconstruction attack. We apply this reasoning to instance encoding and show that dFIL can serve as a good measure for the invertibility of instance encoding, as it can bound the reconstruction MSE against any attacker that tries to reconstruct the private input from the encoding. dFIL can be easily calculated for any encoders that are differentiable and randomized—being applicable to most of the popular encoders with minimal modifications. To the best of our knowledge, our work is the first to theoretically lower-bound the invertibility of instance encoding and use it to design private training/inference systems.

**Contributions**    Our main contributions are as follows:

1. We adapt the result of prior works [18, 20] for instance encoding to show how dFIL can lower bound the MSE of *particular* input reconstruction attacks (*i.e.*, *unbiased* attacks) that aim to reconstruct the raw data given the encoding. We show how popular encoders can be modified minimally for dFIL to be applied (Section 3.1).
2. We extend the result of prior works [18, 20] and show that dFIL can lower bound the MSE of *any* input reconstruction attack (*e.g.*, strong attacks leveraging knowledge of the input prior; Section 3.2). Our extension involves a novel application of the classical *van Trees inequality* [62] and connecting it to the problem of *score matching* [57] in distribution estimation.
3. We evaluate the lower bound using different attacks and encoders, and show that dFIL can be used to interpret the invertibility of instance encoding both theoretically and empirically (Section 3.3).
4. We show how dFIL can be used as a practical privacy metric and guide the design of privacy-enhancing training/inference systems with instance encoding (Section 4–5). We show that it is possible to achieve both high (theoretically-justified) privacy and satisfactory utility.

## 2    Motivation and background

### 2.1    Instance encoding

Instance encoding refers to the general concept of encoding raw input $\mathbf{x}$ using an encoding function $\mathrm{Enc}$, such that private information contained in $\mathbf{x}$ cannot be inferred from its encoding $\mathbf{e} = \mathrm{Enc}(\mathbf{x})$, while $\mathbf{e}$ maintains enough utility for the downstream ML tasks. The principle behind the privacy-enhancing property of instance encoding is that the function $\mathrm{Enc}$ is hard to invert. However, prior works generally justify this claim of non-invertibility based on heuristics rather than rigorous theoretical analysis [64, 65, 45], except for a few cases where the encoder is very simple [16, 17, 44].

**Attacks against instance encoding**    Given an instance encoder $\mathrm{Enc}$, the goal of a reconstruction attack is to recover its input. Formally, given $\mathbf{e} = \mathrm{Enc}(\mathbf{x})$, an attack $\mathrm{Att}$ aims to reconstruct $\mathbf{x}$ from $\mathbf{e}$: $\hat{\mathbf{x}} = \mathrm{Att}(\mathbf{e})$. Such an attack can be carried out in several ways. If $\mathrm{Enc}$ is known, $\hat{\mathbf{x}}$ can be obtained by solving the following optimization [27]: $\hat{\mathbf{x}} = \arg\min_{\mathbf{x_0}}||\mathbf{e} - \mathrm{Enc}(\mathbf{x_0})||_2^2$. This attack can be further improved when some prior of the input is known [49, 61]. For instance, images can be regularized with total variation (TV) prior to reduce high-frequency components [49]. Alternatively, if samples from the input distribution can be obtained, a DNN that generates $\hat{\mathbf{x}}$ from $\mathbf{e}$ can be trained [53, 27, 12].

**Privacy metrics for instance encoding** *Measuring how much information an observation leaks about a secret quantity it depends on* is a classical question that has long been studied in information theory [38, 3, 13, 33], and instance encoding privacy is directly relevant to the stream of works. Prior works proposed metrics providing a strong notion of privacy (*e.g.*, Arimoto's mutual information [3] or Sibson's mutual information [38]). However, it is challenging to apply these metrics to ML and instance encoding which use high-dimensional data [33], because these metrics have to be calculated and optimized over the prior of the dataset, the encodings, and/or their joint distribution, all of which are unknown and hard to accurately model [33].

Differential privacy (DP) [14, 15], one of the most popular frameworks to quantify ML privacy [1], is not suitable for instance encoding [6]. DP guarantees the worst-case indistinguishability of the encoding from two different inputs, which significantly damages the utility [6] (Appendix 7.5). Metric-DP [8]—a weaker notion of DP—is applicable to instance encoding and has been used by several prior works [16, 17, 44, 48, 37]. However, metric-DP is applicable only when the encoder is extremely simple. For example, prior works used pixel-level perturbation (*e.g.*, pixelization, blurring [16, 17]) or a single-layer convolution [44] as an encoder. These extremely-simple encoders are in stark contrast with the complex encoders most practitioners use (*e.g.*, split learning [64], split inference [65], and vFL [45, 59] all uses DNN-based encoders), and results in a significant utility loss (Section 4.2). Applying metric-DP to an arbitrary DNN-based encoder is NP-hard [66]. Unlike these works that focus on indistinguishability, our work directly quantifies *non-invertibility*, a weaker but still relevant definition of privacy (Section 3).

Due to the limitations in existing theoretical metrics, the vast majority of works in the field are relying on empirical measures to demonstrate their encoder's privacy-enhancing properties. Some [46, 45] simply test their encoders against a limited set of attacks, while others [64, 65, 51] use heuristical privacy metrics without rigorous theoretical arguments. Using heuristical metrics that are not backed by a theoretical interpretation is dangerous, because a seemingly useful metric can actually severely mis-characterize the privacy of a system [38]. It is of both interest and practical importance to develop a privacy metric that is both theoretically-meaningful and widely applicable.

A concurrent work [72] proposed a notion of PAC privacy that can bound the adversary's attack success rate in theory for instance encoding. PAC privacy bound has not yet been evaluated empirically under a realistic attack and use cases, which remains an interesting future work.

## 2.2 Fisher information leakage

Fisher information leakage (FIL; [18, 22, 20]) is a measure of leakage through a privacy-enhancing mechanism. Let $\mathcal{M}$ be a randomized mechanism on data sample $\mathbf{x}$, and let $\mathbf{o} \sim \mathcal{M}(\mathbf{x})$ be its output. Suppose the log density function $\log p(\mathbf{o}; \mathbf{x})$ is differentiable w.r.t. $\mathbf{x}$ and satisfies the following regularity condition: $\mathbb{E}_{\mathbf{o}}\left[\nabla_{\mathbf{x}} \log p(\mathbf{o}; \mathbf{x}) \mid \mathbf{x}\right] = 0$. The *Fisher information matrix* (FIM) $\mathcal{I}_{\mathbf{o}}(\mathbf{x})$ is:

$$\mathcal{I}_{\mathbf{o}}(\mathbf{x}) = \mathbb{E}_{\mathbf{o}}[\nabla_{\mathbf{x}} \log p(\mathbf{o}; \mathbf{x}) \nabla_{\mathbf{x}} \log p(\mathbf{o}; \mathbf{x})^{\top}]. \tag{1}$$

**Cramér-Rao bound** Fisher information is a compelling privacy metric as it directly relates to the mean squared error (MSE) of a reconstruction adversary through the Cramér-Rao bound [41]. In detail, suppose that $\hat{\mathbf{x}}(\mathbf{o})$ is an *unbiased* estimate (or reconstruction) of $\mathbf{x}$ given the output of the randomized private mechanism $\mathbf{o} \sim \mathcal{M}(\mathbf{x})$. Then:

$$\mathbb{E}_{\mathbf{o}}[\|\hat{\mathbf{x}}(\mathbf{o}) - \mathbf{x}\|_2^2 / d] \geq \frac{d}{\mathrm{Tr}(\mathcal{I}_{\mathbf{o}}(\mathbf{x}))}, \tag{2}$$

where $d$ is the dimension of $\mathbf{x}$ and $\mathrm{Tr}$ is the trace of a matrix. Guo et al. [20] defined a scalar summary of the FIM called *diagonal Fisher information leakage* (dFIL):

$$\mathrm{dFIL}(\mathbf{x}) = \mathrm{Tr}(\mathcal{I}_{\mathbf{o}}(\mathbf{x}))/d, \tag{3}$$

hence the MSE of an unbiased reconstruction is lower bounded by the reciprocal of dFIL. dFIL varies with $\mathbf{x}$, allowing it to reflect the fact that certain samples may be more vulnerable to reconstruction.

**Limitations** Although the Cramér-Rao bound gives a mathematically rigorous interpretation of dFIL, it depends crucially on the *unbiasedness* assumption, *i.e.*, $\mathbb{E}_{\mathbf{o}}[\hat{\mathbf{x}}(\mathbf{o})] = \mathbf{x}$. In practice, most real-world reconstruction attacks use either implicit or explicit priors about the data distribution and are *biased* (*e.g.*, attacks using TV prior or a DNN). It is unclear how dFIL should be interpreted in these more realistic settings. In Section 3.2, we give an alternative theoretical interpretation based on the van Trees inequality [62], which lower-bounds the MSE of *any* adversary, biased or unbiased.

# 3 Quantifying the invertibility of an encoding

Motivated by the lack of theoretically-principled privacy metrics, we propose to adapt the Fisher information leakage framework to quantify the invertibility of instance encoding. We show that many existing encoders can be modified minimally to be interpreted with dFIL and the Cramér-Rao bound. Subsequently, we extend the framework and derive a novel bound for the reconstruction error of *arbitrary* attacks, by establishing a connection to van Trees inequality and score matching.

**Threat model**   We focus on reconstruction attacks that aim to invert an encoding, *i.e.*, reconstruct the input $\mathbf{x}$ given its encoding $\mathbf{e}$. We assume that the attacker has full knowledge of the encoder $\mathrm{Enc}$ except for the source of randomness. We consider both unbiased attacks and biased attacks that can use arbitrary prior knowledge about the data distribution.

**Privacy definition**   We consider $\mathrm{Enc}$ to be private if $\mathbf{x}$ cannot be reconstructed from the encoding $\mathbf{e}$. We measure the reconstruction error with the mean squared error (MSE), defined as $||\hat{\mathbf{x}} - \mathbf{x}||_2^2/d$. Although MSE does not exactly indicate semantic similarity, it is widely applicable and used as a proxy for semantic similarity [69, 43]. Preventing low reconstruction MSE does not necessarily protect against other attacks (*e.g.*, property inference [50]), which we leave as future work.

## 3.1 Fisher information leakage for instance encoding

To adapt the framework of Fisher information to the setting of instance encoding, we consider the encoding function $\mathrm{Enc}$ as a privacy-enhancing mechanism (*cf.* $\mathcal{M}$ in Section 2.2) and use dFIL to measure the invertibility of the encoding $\mathbf{e} = \mathrm{Enc}(\mathbf{x})$. However, many instance encoders do not meet the regularity conditions in Section 2.2, making dFIL ill-defined. For example, popular DNN-based encoders do not produce randomized output, and their log density function $\log p(\mathbf{o}; \mathbf{x})$ may not be differentiable when operators like ReLU or max pooling are present.

Fortunately, many popular encoders can meet the required conditions with small changes. For example, DNN-based encoders can be modified by (1) replacing any non-smooth functions with smooth functions (*e.g.*, $\tanh$ or GELU [29] instead of ReLU, average pooling instead of max pooling), and (2) adding noise at the end of the encoder for randomness. Alternative modifications are also possible. In particular, if we add random Gaussian noise to the output of a differentiable, deterministic encoder $\mathrm{Enc}_D$ (*e.g.,* DNN): $\mathrm{Enc}(\mathbf{x}) = \mathrm{Enc}_D(\mathbf{x}) + \mathcal{N}(0, \sigma^2)$, the FIM of the encoder becomes [22]:

$$\mathcal{I}_{\mathbf{e}}(\mathbf{x}) = \frac{1}{\sigma^2} \mathbf{J}_{\mathrm{Enc}_D}^\top(\mathbf{x}) \mathbf{J}_{\mathrm{Enc}_D}(\mathbf{x}), \tag{4}$$

where $\mathbf{J}_{\mathrm{Enc}_D}$ is the Jacobian of $\mathrm{Enc}_D$ with respect to the input $\mathbf{x}$ and can be easily computed using a single backward pass. Then, Equation 2 can be used to bound the reconstruction error, *provided the attack is unbiased*. Other popular encoders can be modified similarly.

## 3.2 Bounding the reconstruction of arbitrary attacks

Most realistic reconstruction attacks are biased, and their reconstruction MSE is not lower bounded by Equation 2. Consider an (biased) attacker who knows the mean $\mu$ of the input data distribution. If the attacker simply outputs $\mu$ as the reconstruction of any input $\mathbf{x}$, the expected MSE will be the variance of the data distribution *regardless of dFIL*. The above example shows a crucial limitation of the Cramér-Rao bound interpretation of dFIL: it does not take into account any *prior information* the adversary has about the data distribution, which is abundant in the real world (Section 2.1).

**Bayesian interpretation of Fisher information**   We adopt a Bayesian interpretation of dFIL as the difference between an attacker's prior and posterior estimate of the input $\mathbf{x}$. This is achieved through the classical *van Trees inequality* [62]. We state the van Trees inequality in Appendix 7.3, and use it to derive our MSE bound for arbitrary attacks below as a corollary; proof is in Appendix 7.4.

**Corollary 1.** *Let $\pi$ be the input data distribution and let $f_\pi(\mathbf{x})$ denote its density function with respect to Lebesgue measure. Suppose that $\pi$ satisfies the regularity conditions of van Trees inequality (Theorem 2), and let $\mathcal{J}(f_\pi) = \mathbb{E}_\pi[\nabla_\mathbf{x} \log f_\pi(\mathbf{x}) \nabla_\mathbf{x} \log f_\pi(\mathbf{x})^\top]$ denote the information theorist's Fisher information [2] of $\pi$. For a private mechanism $\mathcal{M}$ and any reconstruction attack $\hat{\mathbf{x}}(\mathbf{o})$*

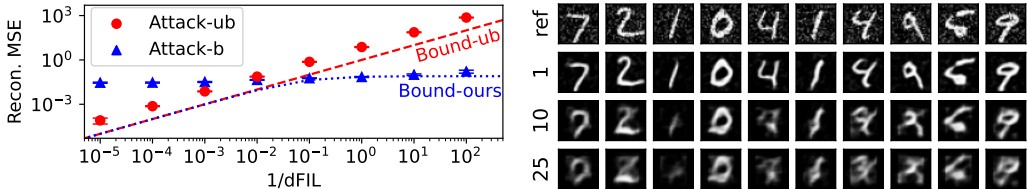

Figure 2: $1/\mathrm{dFIL}$ vs. reconstruction MSE (left) and sample reconstructions (right) for MNIST. The values before each row (right) indicate $1/\mathrm{dFIL}$ of the encoder used.

*operating on* $\mathbf{o} \sim \mathcal{M}(\mathbf{x})$:

$$\mathbb{E}_\pi \mathbb{E}[||\hat{\mathbf{x}} - \mathbf{x}||_2^2/d] \geq \frac{1}{\mathbb{E}_\pi[\mathrm{dFIL}(\mathbf{x})] + \mathrm{Tr}(\mathcal{J}(f_\pi))/d}. \qquad (5)$$

**Implications of Corollary 1**    We can readily apply Corollary 1 to instance encoding by replacing $\mathcal{M}$ with $\mathrm{Enc}$ and $\mathbf{o}$ with $\mathbf{e}$ as in Section 3.1. Doing so leads to several interesting implications:

1. Corollary 1 is a population-level bound that takes expectation over $\mathbf{x} \sim \pi$. This is necessary because given any *fixed* sample $\mathbf{x}$, there is always an attack $\hat{\mathbf{x}}(\mathbf{e}) = \mathbf{x}$ that perfectly reconstructs $\mathbf{x}$ without observing the encoding $\mathbf{e}$. Such an attack would fail in expectation over $\mathbf{x} \sim \pi$.

2. The term $\mathcal{J}(f_\pi)$ captures the prior knowledge about the input. When $\mathcal{J}(f_\pi) = 0$, the attacker has no prior information about $\mathbf{x}$, and Corollary 1 reduces to the unbiased bound in Equation 2. When $\mathcal{J}(f_\pi)$ is large, the bound becomes small regardless of $\mathbb{E}_\pi[\mathrm{dFIL}(\mathbf{x})]$, indicating that the attacker can simply guess with the input prior and achieve a low MSE.

3. dFIL can be interpreted as capturing how much *easier* reconstructing the input becomes after observing the encoding ($\mathbb{E}_\pi[\mathrm{dFIL}(\mathbf{x})]$ term) as opposed to only having knowledge of the input distribution ($\mathrm{Tr}(\mathcal{J}(f_\pi))/d$ term). Using such a *change in the belief after an observation* as a privacy metric is common in information theory [38, 13].

**Estimating $\mathcal{J}(f_\pi)$**    The term $\mathcal{J}(f_\pi)$ captures the prior knowledge of the input and plays a crucial role in Corollary 1. In simple cases where $\pi$ is a known distribution whose density function follows a tractable form (*e.g.*, when the input follows a Gaussian distribution), $\mathcal{J}(f_\pi)$ can be directly calculated. In such settings, Corollary 1 gives a meaningful theoretical lower bound for the reconstruction MSE. However, most real-world data distributions do not have a tractable form and $\mathcal{J}(f_\pi)$ must be estimated from data. Fortunately, the $\nabla_\mathbf{x} \log f_\pi(\mathbf{x})$ term in $\mathcal{J}(f_\pi)$ is a well-known quantity called the *score function* in generative AI literature, and there exists a class of algorithms known as *score matching* [36, 47, 57] that aim to estimate the score function from the data. We leverage these techniques to estimate $\mathcal{J}(f_\pi)$ when it cannot be calculated; details are in Appendix 7.1. As $\mathcal{J}(f_\pi)$ only depends on the dataset and not the encoder, its value can be estimated once for each dataset and shared across the community. The estimation of $\mathcal{J}(f_\pi)$ is expected to improve with the advancement of generative AI.

**Using Corollary 1 in practice**    When $\mathcal{J}(f_\pi)$ is known (*e.g.*, Gaussian), the bound from Corollary 1 always hold. However, when estimating $\mathcal{J}(f_\pi)$ from data, the bound can be incorrect due to several reasons, including improper modeling of the score function, not having enough representative samples, or the regularity conditions of the van Trees inequality not being met. The bound can also be loose when tightness conditions of the van Trees inequality do not hold. Even when the bound is not exact, however, Equations 2 and 5 can be interpreted to suggest that increasing $1/\mathrm{dFIL}$ makes reconstruction harder. Thus, we argue that dFIL still serves as a useful privacy metric that in theory bounds the invertibility of an instance encoding. When not exact, the bound should be viewed more as a *guideline* for interpreting and setting dFIL in a data-dependent manner.

### 3.3    Evaluation of the bound

We show that Corollary 1 accurately reflects the reconstruction MSE on both (1) synthetic data with known $\mathcal{J}(f_\pi)$, and (2) real world data with estimated $\mathcal{J}(f_\pi)$.

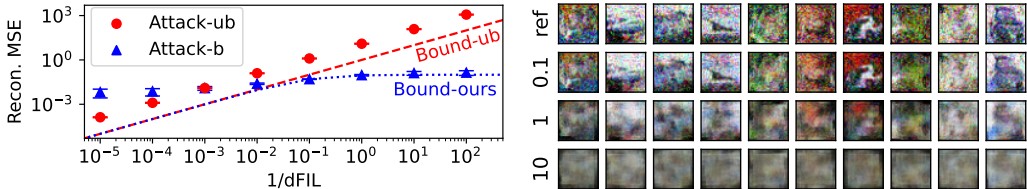

Figure 3: $1/\,\mathrm{dFIL}$ vs. reconstruction MSE (left) and sample reconstructions (right) for CIFAR-10. The values before each row (right) indicate $1/\,\mathrm{dFIL}$ of the encoder used.

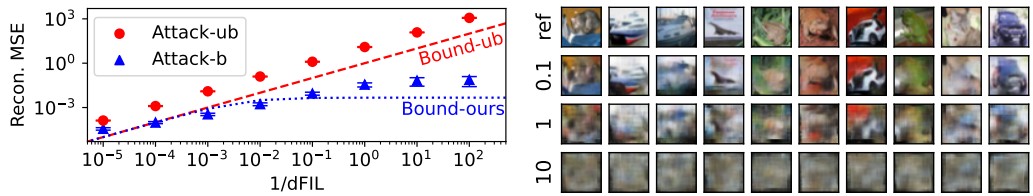

Figure 4: $1/\,\mathrm{dFIL}$ vs. reconstruction MSE (left) and sample reconstructions (right) for CIFAR-10, with a randomized smoothing noise of $\mathcal{N}(0, 0.01^2)$. Corollary 1 breaks when the smoothing noise is too small, but dFIL still shows a strong correlation with the reconstruction quality. The values before each row (right) indicate $1/\,\mathrm{dFIL}$ of the encoder used.

### 3.3.1 Synthetic data with known $\mathcal{J}(f_\pi)$

**Evaluation setup** We consider a synthetic Gaussian input distribution: $\mathbf{x} \sim \mathcal{N}(0, \tau^2 \mathbf{I}_d)$ with $d = 784$ and $\tau = 0.05$. It can be shown that $\mathrm{Tr}(\mathcal{J}(f_\pi))/d = 1/\tau^2$, hence a larger $\tau$ forces the data to spread out more and reduces the input prior. We use a simple encoder which randomly projects the data to a $10{,}000$-dimensional spaces and then adds Gaussian noise, *i.e.*, $\mathbf{e} = \mathbf{M}\mathbf{x} + \mathcal{N}(0, \sigma^2)$, where $\mathbf{M} \in \mathbb{R}^{10{,}000 \times 784}$. We evaluate with a simple encoder to make the attack easier, but note that our bound should hold for more complex encoders (*e.g.*, DNN-based) as well.

**Attacks** We evaluate our bound against two different attacks. An unbiased attack (*Attack-ub*) solves the following optimization: $\hat{\mathbf{x}}(\mathbf{e}) = \arg\min_{\mathbf{x_0}} ||\mathbf{e} - \mathrm{Enc}(\mathbf{x_0})||_2^2$. The attack is unbiased as the objective is convex, and $\mathbf{x}$ is recovered in expectation. A more powerful biased attack (*Attack-b*) adds a regularizer term $\lambda \log p_\tau(\mathbf{x_0})$ to the above objective, where $p_\tau$ is the density function of $\mathcal{N}(0, \tau^2 \mathbf{I}_d)$. One can show that with a suitable choice of $\lambda$, this attack returns the *maximum a posteriori* estimate of $\mathbf{x}$, which leverages knowledge of the input distribution. Details are in Appendix 7.2.

**Result** Figure 1 (right) plots the MSE of the two attacks, and the bounds for unbiased (Equation 2) and arbitrary attack (Equation 5). The MSE of *Attack-ub* (red circle) matches the unbiased attack lower bound (*Bound-ub*; red dashed line), showing the predictive power of Equation 2 against this restricted class of attacks. Under *Attack-b* (blue triangle), however, *Bound-ub* breaks. Our new bound from Equation 5 (*Bound-ours*; blue dotted line) reliably holds for both attacks, initially being close to the unbiased bound and converging to guessing only with the input prior (attaining $\tau^2$).

### 3.3.2 Real world data with estimated $\mathcal{J}(f_\pi)$

**Evaluation setup** We also evaluated Corollary 1 on MNIST [10] and CIFAR-10 [42]. Here, we estimated $\mathcal{J}(f_\pi)$ using sliced score matching [57]. As discussed in Appendix 7.1, a moderate amount of randomized smoothing (adding Gaussian noise to the raw input [9]) is necessary to ensure that the score estimation is stable and that regularity conditions of the van Trees inequality are satisfied. We used a simple CNN-based encoder: $\mathbf{e} = \mathrm{Conv}(\mathbf{x}) + \mathcal{N}(0, \sigma^2)$.

**Attacks** We evaluated *Attack-ub*, which is the same as in Section 3.3.1, and *Attack-b*, which is a trained DNN that outputs the reconstruction given an encoding [45]. We also evaluated regularization-based attacks [49, 61] and obtained similar results; we omit those results for brevity.

**Result**   Figures 2 and 3 plot the result with a randomized smoothing noise of $\mathcal{N}(0, 0.25^2)$. Again, *Bound-ub* correctly bounds the MSE achieved by *Attack-ub*. While *Attack-b* is not as effective for very low $1/\,\mathrm{dFIL}$, it outperforms *Attack-ub* for high $1/\,\mathrm{dFIL}$, breaking *Bound-ub*. In comparison, Corollary 1 estimated using score matching (*Bound-ours*) gives a valid lower bound for both attacks.

Figures 2 and 3 also highlights some of the reconstructions visually. Here, the left-hand side number indicates the target $1/\,\mathrm{dFIL}$ and the images are reconstructed using *Attack-b*. In both figures, it can be seen that dFIL correlates well with the visual quality of reconstructed images, with higher values of $1/\,\mathrm{dFIL}$ indicating less faithful reconstructions. See Appendix: Figure 10–11 for more results.

Figure 4 additionally shows the result with a much smaller randomized smoothing noise of $\mathcal{N}(0, 0.01^2)$. Unlike previous results, *Bound-ours* breaks around $1/\,\mathrm{dFIL}=10^{-3}$. We suspect it is due to score matching failing when the data lie on a low-dimensional manifold and the likelihood changes rapidly near the manifold boundary, which can be the case when the smoothing noise is small. Nonetheless, the bound still correlates well with MSE and the visual reconstruction quality. We claim that dFIL still serves as a useful privacy metric, with a theoretically-principled interpretation and a strong empirical correlation to invertibility. More reconstructions are in Appendix: Figure 12.

## 4   Case study 1: split inference with dFIL

### 4.1   Private split inference with dFIL

Split inference [40, 4, 65] is a method to run inference of a large DNN that is hosted on the server, without the client disclosing raw input. It is done by running the first few layers of a large DNN on the client device and sending the intermediate activation, instead of raw data, to the server to complete the inference. The client computation can be viewed as instance encoding, where the first few layers on the client device act as an encoder. However, without additional intervention, split inference by itself is *not* private because the encoding can be inverted [27].

We design a private split inference system by measuring and controlling the invertibility of the encoder with dFIL. Because the encoder of split inference is a DNN, dFIL can be calculated using Equation 4 with minor modifications to the network (Section 3.1).

**Optimizations**   There are several optimizations to improve the model accuracy for the same dFIL: (1) We calculate the amount of noise that needs to be added to the encoding to achieve a target dFIL, and add a similar amount of noise during training. (2) For CNNs, we add a compression layer—a convolution layer that reduces the channel dimension significantly—at the end of the encoder and a corresponding decompression layer at the beginning of the server-side model. Similar heuristics were explored in prior works [11, 45]. (3) We add an *SNR regularizer* that is designed to maximize the signal-to-noise ratio of the encoding. From Equations 3–4, the noise that needs to be added to achieve a certain dFIL is $\sigma = \sqrt{\frac{\mathrm{Tr}(\mathbf{J}_{\mathrm{Enc}_D}^\top(\mathbf{x})\mathbf{J}_{\mathrm{Enc}_D}(\mathbf{x}))}{d*\mathrm{dFIL}}}$. Thus, maximizing the signal-to-noise ratio (SNR) of the encoding ($\mathbf{e}^\top \mathbf{e}/\sigma^2$) is equivalent to minimizing $\frac{\mathrm{Tr}(\mathbf{J}_{\mathrm{Enc}_D}^\top(\mathbf{x})\mathbf{J}_{\mathrm{Enc}_D}(\mathbf{x}))}{\mathbf{e}^\top \mathbf{e}}$, which we add to the optimizer during training. These optimizations were selected from comparing multiple heuristics from prior work [60, 28, 46, 65, 45, 11], and result in a notable improvement of test accuracy.

### 4.2   Evaluation of dFIL-based split inference

#### 4.2.1   Evaluation setup

**Models and datasets**   We used three different models and four different datasets to cover a wide range of applications: ResNet-18 [25] with CIFAR-10/100 [42] for image classification, MLP-based neural collaborative filtering (NCF-MLP) [26] with MovieLens-20M [24] for recommendation, and DistilBert [56] with GLUE-SST2 [68] for sentiment analysis. See Appendix 7.2 for details.

**Detailed setups and attacks**   For ResNet-18, we explored three split inference configurations: splitting early (after the first convolution layer), in the middle (after block 4), and late (after block 6). We evaluated the empirical privacy with a DNN attacker [45] and measured the reconstruction quality with structural similarity index measure (SSIM) [31]. Other popular attacks showed similar trends (see Appendix: Figure 9). NCF-MLP translates a user id (uid) and a movie id (mid) into embeddings

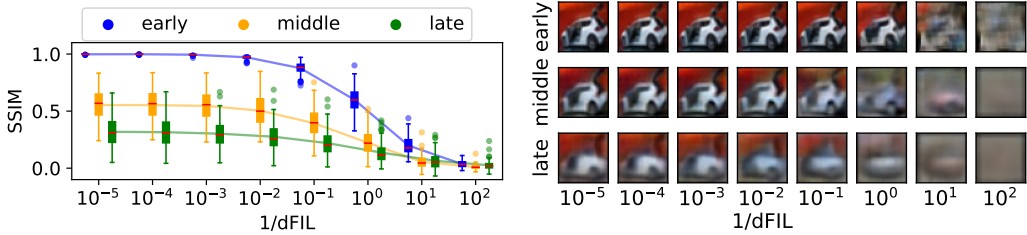

Figure 5: $1/\mathrm{dFIL}$ vs. SSIM (left) and sample reconstructions (right) for split inference with ResNet-18 and CIFAR-10. Higher SSIM indicates more successful reconstruction.

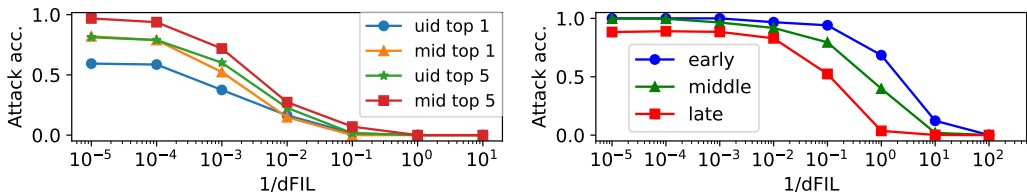

Figure 6: Attack accuracy vs. $1/\mathrm{dFIL}$ for split inference with NCF-MLP (left) and DistilBert (right).

with an embedding table and sends them through a DNN to make a prediction. We split the NCF-MLP model after the first linear layer of the MLP and tried reconstructing the original uid and mid from the encoding. This is done by first reconstructing the embeddings from the encoding using direct optimization $(\hat{emb}(\mathbf{e}) = \arg\min_{emb_0}||\mathbf{e} - \mathrm{Enc}(emb_0)||_2^2)$, and finding the original uid and mid by choosing the closest embedding value in the embedding table: $id = \arg\min_{i}||\hat{emb} - Emb[i]||_2^2$, where $Emb[i]$ is the $i$-th entry of the embedding table. For DistilBert, we again explored three different splitting configurations: splitting early (right after block 0), in the middle (after block 2), and late (after block 4). We use a similar attack to NCF-NLP to retrieve each word token.

### 4.2.2 Evaluation results

**Privacy** Figure 5 shows the attack result for ResNet-18 and CIFAR-10. Setups with lower dFIL lead to lower SSIM (left) and less identifiable images (right), indicating that dFIL strongly correlates with the attack success rate. The figures also show that the privacy leakage estimated by dFIL can be conservative. Some setups show empirically-high privacy even when dFIL indicates otherwise, especially when splitting late. See Appendix: Figure 13 for more results. Figures 6 show the attack result for NCF-MLP (left) and DistilBert (right). Setups with lower dFIL again consistently showed a worse attack success rate. A sample reconstruction for DistilBert is shown in Appendix: Table 5.

**Utility** Table 1 summarizes the test accuracy of the split inference models, where $1/\mathrm{dFIL}$ is chosen so that the attacker's reconstruction error is relatively high. For the same $1/\mathrm{dFIL}$, our proposed optimizations (**Ours**) improve the accuracy significantly compared to simply adding noise (**No opt.**). In general, reasonable accuracy can be achieved with encoders with relatively low dFIL. Accuracy degrades more when splitting earlier, indicating that using a DNN encoder with many layers can benefit utility. Prior works empirically observed a similar trend where splitting earlier degrades the privacy-utility tradeoff, because the encoding becomes leakier [49]. Our framework supports complex DNN encoders, which is a distinguishing benefit compared to prior theoretical works [16, 17, 44].

## 5 Case study 2: training with dFIL

### 5.1 Training on encoded data with dFIL

We consider a scenario where users publish their encoded private data, and a downstream model is trained on the encoded data. We use the first few layers of a pretrained model as the encoder by

Table 1: Test accuracy for different split inference setups with different dFIL. Base accuracy in parenthesis.

| Setup | Split | $\frac{1}{dFIL}$ | No opt. | Ours |
|---|---|---|---|---|
| CIFAR-10 + ResNet-18 (acc: 92.70%) | early | 10 | 10.70% | **74.44%** |
| | | 100 | 10.14% | **57.97%** |
| | middle | 10 | 22.11% | **91.35%** |
| | | 100 | 12.94% | **84.27%** |
| | late | 10 | 78.48% | **92.35%** |
| | | 100 | 33.54% | **87.58%** |
| CIFAR-100 + ResNet-18 (acc: 72.13%) | early | 10 | 1.00% | **44.10%** |
| | | 100 | 1.00% | **30.69%** |
| | middle | 10 | 1.98% | **59.51%** |
| | | 100 | 1.29% | **41.29%** |
| | late | 10 | 8.63% | **65.77%** |
| | | 100 | 2.01% | **39.18%** |
| MovieLens-20M + NCF-MLP (AUC: 0.8228) | early | 1 | 0.8172 | **0.8286** |
| | | 10 | 0.7459 | **0.8251** |
| | | 100 | 0.6120 | **0.8081** |
| GLUE-SST2 + DistilBert (acc: 91.04%) | early | 10 | 50.80% | **82.80%** |
| | | 100 | 49.08% | **81.88%** |
| | middle | 10 | 76.61% | **83.03%** |
| | | 100 | 61.93% | **82.22%** |
| | late | 10 | **90.25%** | 83.03% |
| | | 100 | 82.68% | **82.82%** |

Table 2: Accuracy from training with different encoders.

| Pretrain dataset | $\frac{1}{dFIL}$ | Acc. |
|---|---|---|
| TinyImageNet | 10 | 75.46% |
| | 100 | 42.57% |
| CIFAR-100 | 10 | 80.16% |
| | 100 | 70.27% |
| CIFAR-10 (held-out 20%) | 10 | 81.99% |
| | 100 | 78.65% |

freezing the weights and applying the necessary changes in Section 3.1. Then, we use the rest of the model with its last layer modified for the downstream task and finetune it with the encoded data. We found that similar optimizations from split inference benefit this use case as well.

## 5.2 Evaluation of dFIL-based training

We evaluate the model utility and show that it can reach a reasonable accuracy when trained on encoded data. We omit the privacy evaluation as it is similar to Section 4.2.2.

**Evaluation setup**    We train a ResNet-18 model for CIFAR-10 classification. First, we train the model on one of three different datasets: (1) TinyImageNet [58], (2) CIFAR-100 [42], and (3) held-out 20% of CIFAR-10. Then, layers up to block 4 are frozen and used as the encoder. The CIFAR-10 training set is encoded using the encoder and used to finetune the rest of the model. The setup mimics a scenario where some publicly-available data whose distribution is similar (TinyImageNet, CIFAR-100) or the same (held-out CIFAR-10) with the target data is available and is used for encoder training. Detailed hyperparameters are in Appendix 7.2.

**Evaluation result**    Table 2 summarizes the result. Our design was able to achieve a decent accuracy using encoded data with relatively safe $1/dFIL$ values (10–100). The result indicates that model training with privately encoded data is possible. The achieved accuracy was higher when the encoder was trained with data whose distribution is more similar to the downstream task (CIFAR-10). We believe more studies in hyperparameter/architecture search will improve the result.

## 6 Limitations

We showed that dFIL can theoretically quantify the invertibility of instance encoding by providing a reconstruction error bound against arbitrary attackers. We subsequently showed that dFIL can be used to guide private training/inference systems that uses instance encoding. dFIL has several potential limitations, which needs to be carefully considered before being used:

1. Corollary 1 only bounds the MSE, which might not always correlate well with the semantic quality. To address this, van Trees inequality can be extended to an absolutely continuous function $\psi(\mathbf{x})$ to bound $\mathbb{E}[||\psi(\hat{\mathbf{x}}) - \psi(\mathbf{x})||_2^2/d]$ [19], which may be used to extend to metrics other than MSE.

2. Equation 5 provides an average case bound, so some data may experience lower reconstruction MSE than the bound. Similar average case bounds have also been used in prior works [13, 21]. When bounding invertibility, it is unclear how to define the privacy attack game to target worst-case reconstruction bound: for any sample, there exists a trivial attacker that always outputs that fixed sample from the distribution, which gives a worst-case MSE of zero for that sample. Leakier data that might experience lower reconstruction MSE than the bound can be potentially detected and handled by dynamically calculating dFIL for each sample. See Appendix: Figure 14.

3. For data types where MSE is not directly meaningful or the bound is inaccurate, it may not be straightforward to interpret the privacy of an encoding given its dFIL. In such cases, acceptable values of dFIL should be determined for each application through further research. The situation is similar to DP, where it is often not straightforward what privacy parameters (*e.g.*, $\epsilon$, $\delta$) need to be used [39].

4. Systems with the same dFIL may actually have different invertibility, as the bound from dFIL may be conservative. Comparing the privacy of two different systems using dFIL should be done with caution because dFIL is a lower bound rather than an accurate measure of invertibility.

## Acknowledgments and Disclosure of Funding

We thank Yang Song for the invaluable discussion regarding the application of sliced score matching to our setup. We also thank Minjae Park and Jisu Kim for the discussion of our bound's implications. Kiwan Maeng was partially funded by the Charles K. Etner Early Career Professorship from Penn State University.

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

# 7 Appendix

## 7.1 Score matching details

We found that using score matching [57] does not work reliably when the data's structure lies on a low-dimensional manifold (*e.g.*, natural images). We found that applying randomized smoothing [9], which adds Gaussian noise to the image for robust training, helps stabilize score matching as it smoothens the density function. Randomized smoothing also makes the bound tighter. We observed that adding a reasonable amount of noise (*e.g.,* standard deviation of 0.25, which was originally used by Cohen et al. [9]) works well in general, but adding only small noise (standard deviation of 0.01) does not. We show both results in Section 3.3.1.

## 7.2 Hyperparameters

**Attacks** For attacks in Section 3.3.1, 3.3.2, and 4.2, we used the following hyperparameters. For the optimizer-based attack for Gaussian synthetic input, we used Adam with lr=$10^{-3}$, and $\lambda$=0.1–100 for the regularizer. For the optimizer-based attack for NCF-MLP and DistilBert, we used Adam with lr=0.1. For the DNN-based attack for MNIST and CIFAR-10 (Figure 2, 3, 5), we used a modified DNN from Li et al. [45], which uses a series of convolution (Conv) and convolution transpose (ConvT) layers interspersed with leaky ReLU of slope 0.2. All the models were trained for 100 epochs using Adam with lr=$10^{-3}$. Below summarizes the architecture parameters. For DNN-based attacks in Section 3.3.2, we put a sigmoid at the end. For the attack in Section 4.2, we do not.

Table 3: DNN attacker architectures used in the paper. Output channel dimension ($c_{out}$), kernel size (k), stride (s), and output padding (op) are specified. Input padding was 1 for all layers.

| Dataset + encoder | Architecture |
|---|---|
| MNIST + Conv | 3×Conv($c_{out}$=16, k=3, s=1) + ConvT($c_{out}$=32, k=3, s=1, op=0) + ConvT($c_{out}$=1, k=3, s=1, op=0) |
| CIFAR-10 + split-early | 3×Conv($c_{out}$=64, k=3, s=1) + ConvT($c_{out}$=128, k=3, s=1, op=0) + ConvT($c_{out}$=3, k=3, s=1, op=0) |
| CIFAR-10 + split-middle | 3×Conv($c_{out}$=128, k=3, s=1) + ConvT($c_{out}$=128, k=3, s=2, op=1) + ConvT($c_{out}$=3, k=3, s=2, op=1) |
| CIFAR-10 + split-late | 3×Conv($c_{out}$=256, k=3, s=1) + 2×ConvT($c_{out}$=256, k=3, s=2, op=1) + ConvT($c_{out}$=3, k=3, s=2, op=1) |

**Split inference** Below are the hyperparameters for the models used in Section 4.2. For ResNet-18, we used an implementation tuned for CIFAR-10 dataset from [54], with ReLU replaced with GELU and max pooling replaced with average pooling. We used the default hyperparameters from the repository except for the following: bs=128, lr=0.1, and weight_decay=$5 \times 10^{-4}$. For NCF-MLP, we used an embedding dimension of 32 and MLP layers of output size [64, 32, 16, 1]. We trained NCF-MLP with Nesterov SGD with momentum=0.9, lr=0.1, and batch size of 128 for a single epoch. We assumed 5-star ratings as click and others as non-click. For DistilBert, we used Adam optimizer with a batch size of 16, lr=$2 \times 10^{-5}$, $\beta_1$=0.9, $\beta_2$=0.999, and $\epsilon = 10^{-8}$. We swept the compression layer channel dimension among 2, 4, 8, 16, and the SNR regularizer $\lambda$ between $10^{-3}$ and 100.

**Training** Below are the hyperparameters for the models evaluated in Section 5.2. We used the same model and hyperparameters with split inference for training the encoder with the pretraining dataset. Then, we freeze the layers up to block 4 and trained the rest for 10 epochs with CIFAR-10, with lr=$10^{-3}$ and keeping other hyperparameters the same.

**Execution Environment** All the evaluation was done on a single A100 GPU. The training and evaluation of each model ranged roughly from less than an hour (ResNet-18 with split-early, NCF-MLP) to 3–7 hours (ResNet-18 with split-late, DistilBert).

## 7.3 van Trees inequality

Below, we restate the van Trees Inequality [19], which we use to prove Theorem 5.

**Theorem 2** (Multivariate van Trees inequality). *Let $(\mathcal{X}, \mathcal{F}, P_\theta : \theta \in \Theta)$ be a family of distributions on a sample space $\mathcal{X}$ dominated by $\mu$. Let $p(\mathbf{x}|\theta)$ denote the density of $X \sim P_\theta$ and $\mathcal{I}_{\mathbf{x}}(\theta)$ denotes its FIM. Let $\theta \in \Theta$ follows a probability distribution $\pi$ with a density $\lambda_\pi(\theta)$ with respect to Lebesgue measure. Suppose that $\lambda_\pi$ and $p(\mathbf{x}|\theta)$ are absolutely $\mu$-almost surely continuous and $\lambda_\pi$ converges to 0 and the endpoints of $\Theta$. Let $\psi$ be an absolutely continuous function of $\theta$ with the same output dimension of $\theta$, and $\psi_n$ an arbitrary estimator of $\psi(\theta)$. Assume regularity conditions from Section 2.2 is met. If we make $n$ observations $\{\mathbf{x_1}, \mathbf{x_2}, ..., \mathbf{x_n}\}$, then:*

$$\int_\Theta \mathbb{E}_\theta[||\psi_n - \psi(\theta)||_2^2]\lambda_\pi(\theta)d\theta \geq \frac{(\int \operatorname{div} \psi(\theta)\lambda_\pi(\theta)d\theta)^2}{n \int \operatorname{Tr}(\mathcal{I}_{\mathbf{x}}(\theta))\lambda_\pi(\theta)d\theta + \operatorname{Tr}(\mathcal{J}(\lambda_\pi))},$$

*where* $\operatorname{div} \psi(\theta) = \Sigma_i \frac{\partial \psi_i(\theta)}{\partial \theta_i}$.

### 7.4 Proof of Corollary 1

*Proof.* Let $\psi$ be an identity transformation $\psi(\theta) = \theta$. For the setup in Corollary 1, $n = 1$ and $\operatorname{div}(\psi(\theta)) = d$, so the multivariate van Trees inequality from Theorem 2 reduces to:

$$\mathbb{E}_\pi \mathbb{E}_\theta[||\hat{\mathbf{x}} - \mathbf{x}||_2^2/d] \geq \frac{d}{\mathbb{E}_\pi[\operatorname{Tr}(\mathcal{I}_{\mathbf{e}}(\mathbf{x}))] + \operatorname{Tr}(\mathcal{J}(f_\pi))} = \frac{1}{\mathbb{E}_\pi[\operatorname{dFIL}(\mathbf{x})] + \operatorname{Tr}(\mathcal{J}(f_\pi))/d}$$

$\square$

### 7.5 Comparison with differential privacy

Differential privacy [1] is not well-suited for instance encoding, as we discuss in Section 2.1. We formulate and compare a DP-based instance encoding and compare it with our dFIL-based instance encoding in a split inference setup (Section 4) to show that DP-based instance encoding indeed does not work well.

To formulate DP for instance encoding, we define an adjacent set $\mathcal{D}$ and $\mathcal{D}'$ as two differing inputs. A randomized method $\mathcal{A}$ is $(\alpha, \epsilon)$-Rényi differentially private (RDP) if $D_\alpha(\mathcal{A}(\mathcal{D})||\mathcal{A}(\mathcal{D}')) \leq \epsilon$ for $D_\alpha(P||Q) = \frac{1}{\alpha-1} \log \mathbb{E}_{x \sim Q}[(\frac{P(x)}{Q(x)})^\alpha]$. As DP provides a different privacy guarantee with dFIL, we use the theorem from Guo et al. [20] to derive an MSE lower bound using DP's privacy metric for an unbiased attacker. Assuming a reconstruction attack $\hat{\mathbf{x}} = \operatorname{Att}(\mathbf{e})$ that reconstructs $\mathbf{x}$ from the encoding $\mathbf{e} = \operatorname{Enc}(\mathbf{x})$, repurposing the theorem Guo et al. [20] gives:

$$\mathbb{E}[||\hat{\mathbf{x}} - \mathbf{x}||_2^2/d] \geq \frac{\Sigma_{i=1}^d \operatorname{diam}_i(\mathcal{X})^2/4d}{e^\epsilon - 1} \tag{6}$$

for a $(2, \epsilon)$-RDP Enc, where $\mathcal{X}$ is the input data space. We can construct a $(2, \epsilon)$-RDP encoder $\operatorname{Enc}_{RDP}$ from a deterministic encoder $\operatorname{Enc}_D$ by scaling and clipping the encoding adding Gaussian noise, or $\operatorname{Enc}_{RDP} = \operatorname{Enc}_D(\mathbf{x})/\max(1, \frac{||\operatorname{Enc}_D(\mathbf{x})||_2}{C}) + \mathcal{N}(0, \sigma^2)$, similarly to [1]. The noise to be added is $\sigma = \frac{(2C)^2}{\epsilon}$ [52]. Equation 6 for DP is comparable to Equation 2 for dFIL, and we use the two equations to compare DP and dFIL parameters. We use Equation 2 because [20] does not discuss the bound against biased attackers.

We evaluate both encoders for split inference using CIFAR-10 dataset and ResNet-18. We split the model after block 4 (split-middle from Section 4.2.1) and did not add any optimizations discussed in Section 4 for simplicity. For the DP-based encoder, we retrain the encoder with scaling and clipping so that the baseline accuracy without noise does not degrade. We ran both models without standardizing the input, which makes $\operatorname{diam}_i(\mathcal{X}) = 1$ for all $i$.

Table 4 compares the test accuracy achieved when targeting the same MSE bound for an unbiased attacker using dFIL and DP, respectively. The result clearly shows that DP degrades the accuracy much more for similar privacy levels (same unbiased MSE bound), becoming impractical very quickly. DP suffers from low utility because DP is agnostic with the input and the model, assuming a

Table 4: Test accuracy when targeting the same MSE bound.

| Unbiased MSE bound | 1e-5 | 1e-4 | 1e-3 | 1e-2 |
|---|---|---|---|---|
| dFIL-based | **93.09%** | **93.11%** | **92.52%** | **87.52%** |
| DP-based | 64.64% | 56.68% | 46.46% | 33% |

worst-case input and model weights. Our dFIL-based bound uses the information of the input and model weights in its calculation of the bound and can get a tighter bound.

## 7.6 Attack based on a diffusion model

We additionally designed a powerful, diffusion model-based reconstruction attacker to study the privacy of dFIL against the best-effort attacker, motivated from the fact that recently developed diffusion models [30] are excellent denoisers. During training of a diffusion model, (1) a particularly-designed noise is added to an image, and (2) a DNN is trained to predict and remove the noise [30]. The first part can be thought of as an instance encoder (that is purposely made easy to invert), and we can calculate its dFIL. The second part can be thought of as a reconstruction attacker. As the noising and denoising are specifically designed for the denoising to work well, we expect a mature, pretrained diffusion model to give a very good attack quality. We used DDPM [30] pretrained with CIFAR-10 from Google [67].

Figure 7 shows the result. The first column of each row shows the original image, and other columns show the reconstruction of our DDPM-based attacker with different dFIL. We scale and show dFIL with the same scale as Figure 5, as DDPM works with a different normalized image that produces dFIL at a different scale. Our new attack provided an interesting result: the attack was able to reconstruct conceptually similar images with the original image even when pixel-by-pixel reconstruction was prohibited by high $1/\,\mathrm{dFIL}$. For example, 7th row at $1/\,\mathrm{dFIL} = 49.5$ (6th column) successfully reconstructed a white car with a red background, although the reconstruction MSE was high and the design of the reconstructed car was nothing like the original image. The result shows that high-level information of the image (*e.g.*, the color of the car/background, the orientation of the car, *etc.*) can still be preserved after encoding with a relatively high $1/\,\mathrm{dFIL}$, which is why it is possible to perform downstream training/inference with a privately-encoded data without revealing the original data.

## 7.7 Per-pixel FIL

Instead of dFIL which is the average of the Fisher information leakage (FIL) for all the features (pixels), we can directly inspect the per-feature FIL to see which features (pixels) leak more information and can be reconstructed more easily. Figure 8 plots the per-pixel FIL from the split-middle setup of ResNet-18 and CIFAR-10 from Section 4.2.2. The result shows that identifying features of an object, such as the contour of a frog (8th column) or the tire of the car (7th column), are more easily leaked, following our intuition.

## 7.8 Additional figures and tables

Table 5: The reconstruction quality of an input is highly correlated with dFIL. Correct parts in bold.

| 1/dFIL | Reconstructed text (from split-early) |
|---|---|
| $10^{-5}$ | **it's a charming and often affecting journey.** |
| 1 | **it's** cones **charming**ound
**often affecting journey** closure |
| 10 | grounds yuki cum sign
recklessound fanuche pm stunt |

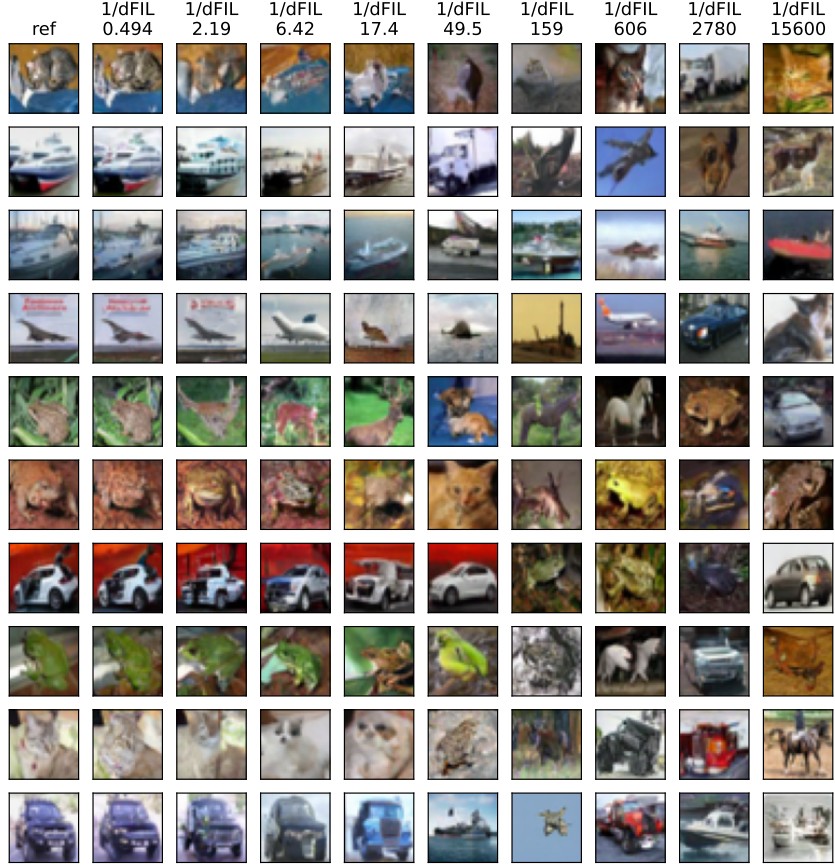

Figure 7: Results from DDPM [30]-based reconstruction attack. High $1/\mathrm{dFIL}$ prevents exact pixel-to-pixel reconstruction, but images that share some high-level features with the original image can be generated unless $1/\mathrm{dFIL}$ is not too high.

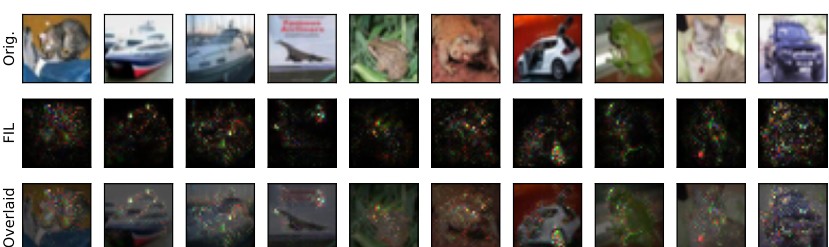

Figure 8: Per-pixel FIL from the ResNet-18 + CIFAR-10 split-middle setup in Section 4.2.2. Image best viewed with colors.

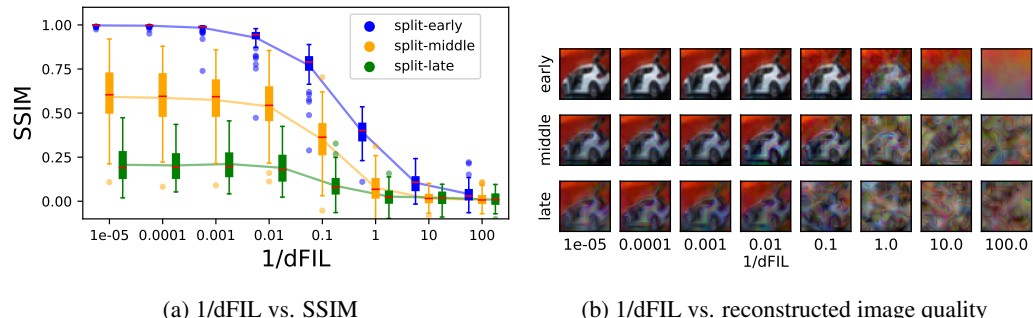

(a) 1/dFIL vs. SSIM

(b) 1/dFIL vs. reconstructed image quality

Figure 9: Optimizer-based attack with total variation (TV) prior [49] against our split inference system in Section 4. The trend is very similar to Figure 5.

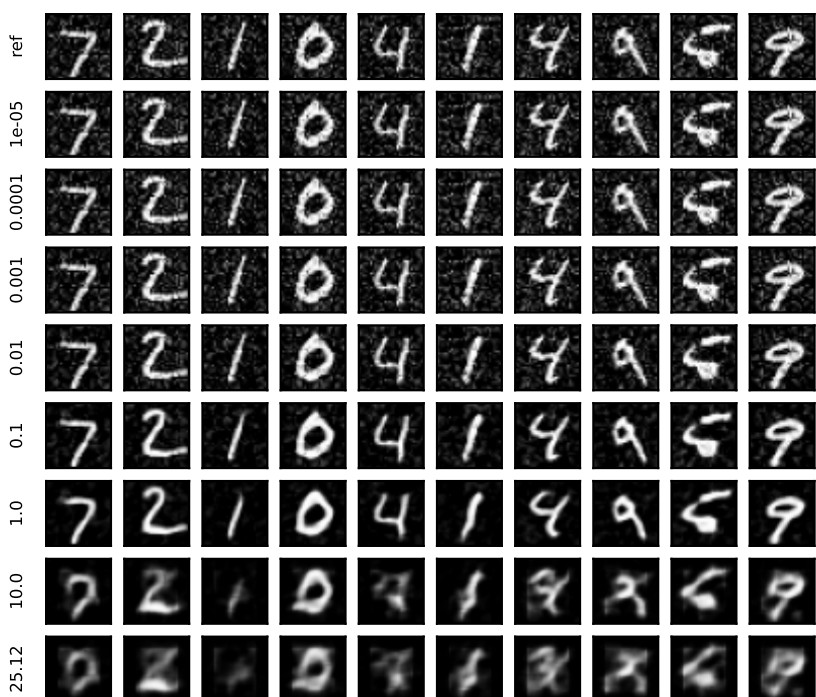

Figure 10: More reconstruction result of Figure 2.

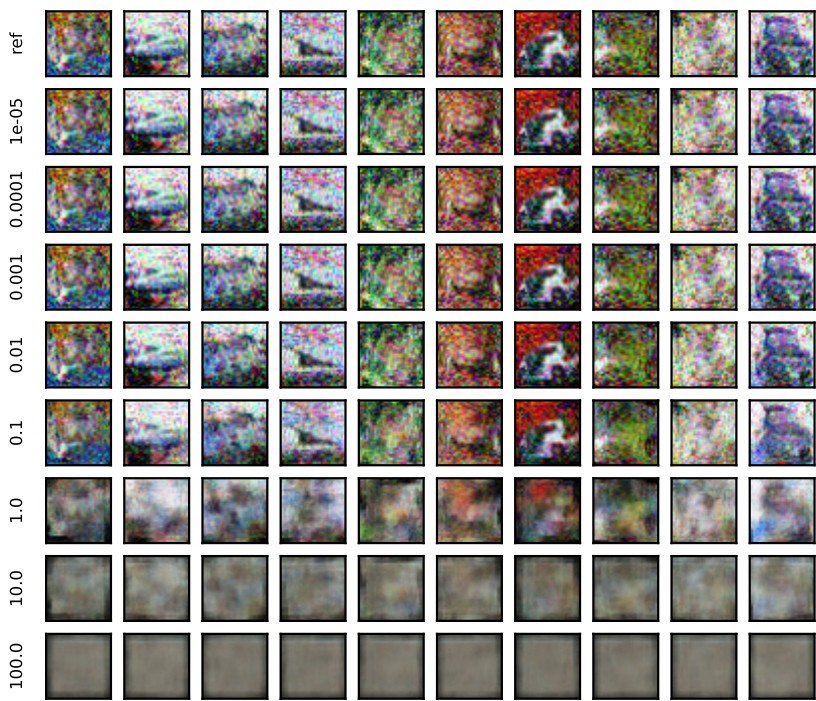

Figure 11: More reconstruction result of Figure 3.

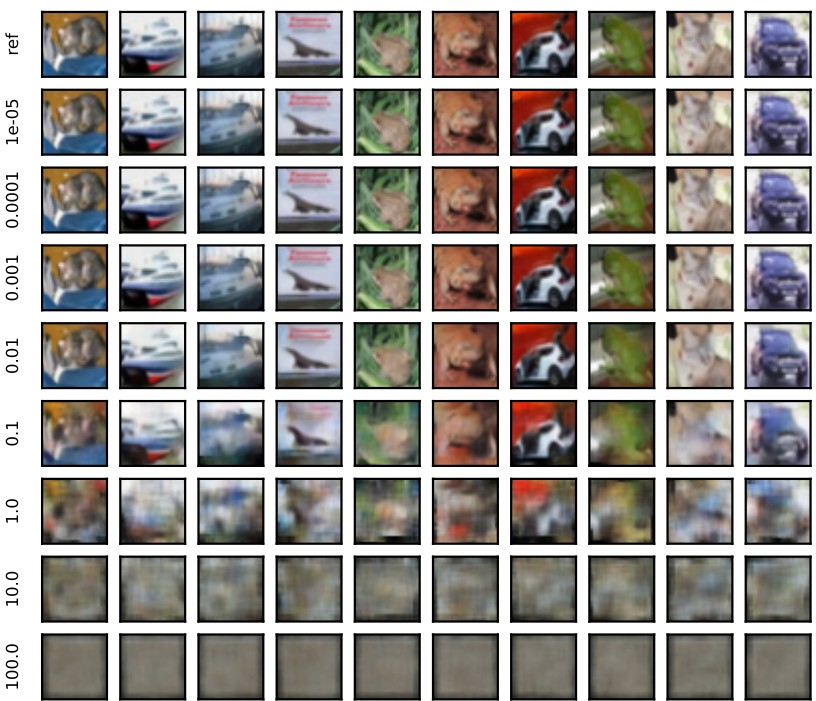

Figure 12: More reconstruction result of Figure 4.

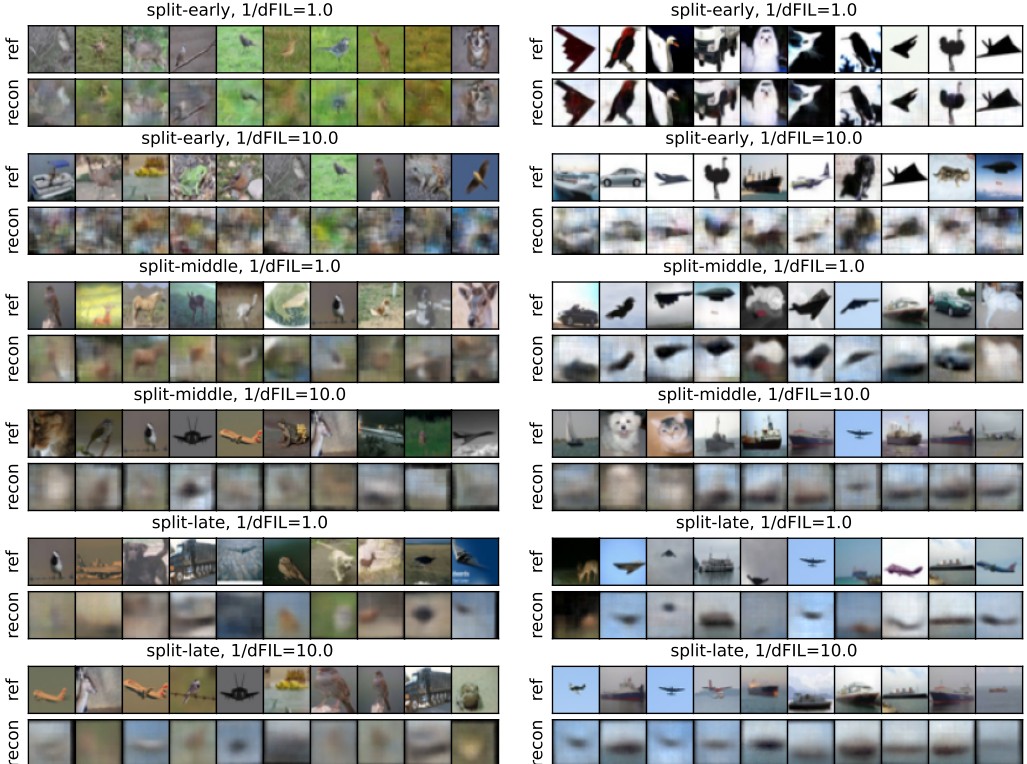

(a) Reconstructions with the worst SSIM values.  (b) Reconstructions with the best SSIM values.

Figure 13: Ten reconstructions with the best and worst SSIM values for various setups. The result is an extension of Figure 5. Images with a simple shape and high color contrast tend to be reconstructed more easily, which matches our intuition. Omitting results from $1/\,\mathrm{dFIL} = 100$, as no meaningful reconstruction was possible.

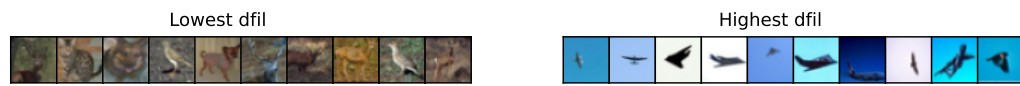

Figure 14: Ten images with the lowest and highest dFIL values, for split-middle setup in Figure 5. Images with high dFIL tend to have a simpler shape and a high color contrast, potentially being easier to reconstruct. Individual dFIL value of each samples can potentially be used to detect data that are more leaky.

