# OpenReview forum: "Bounding the Invertibility of Privacy-preserving Instance Encoding using Fisher Information"
_NeurIPS.cc/2023/Conference — NeurIPS 2023 poster_

### Official Review · Reviewer_AWfY · 2023-06-30

**Soundness:** 4 excellent
**Presentation:** 4 excellent
**Contribution:** 2 fair
**Rating:** 7
**Confidence:** 3

**Summary:**

In this paper the authors propose (diagonal) Fisher information leakage (dFIL) as a theoretically-grounded and practical framework for assessing the privacy guarantees of instance encoding. At its core, dFIL quantifies the potential invertibility of the encoding mapping of an instance encoding scheme. The authors establish, under mild regularity conditions, that the reciprocal of dFIL, plus the information theorist's Fisher information, serves as a lower bound for the mean squared error (MSE) of any input reconstruction attack (Corollary 1). This lower bound is derived from the so-called van Trees' inequality. Furthermore, the authors present extensive numerical experiments that highlight the key features of dFIL as a privacy metric for instance encoding.

**Strengths:**

1. dFIL is a very intuitive privacy metric under unbiased reconstruction attacks. In such cases, the Cramér-Rao bound establishes that the reciprocal of dFIL is a lower bound for the MSE of an adversary attempting to reconstruct the raw data based on the encoding.

2. The limitations of dFIL are discussed thoroughly.

3. The paper is well-written.

**Weaknesses:**

Although dFIL serves as an intuitive privacy metric under unbiased reconstruction attacks, these attacks are uncommon in ML applications. In the case of biased attacks, which are more representative of practical scenarios, the main result (Corollary 1) highlights the need to consider not only dFIL but also the information theorist's Fisher information. Consequently, certain practical and theoretical concerns are shifted to the latter quantity, potentially diminishing the significance of dFIL itself. Furthermore, it is worth mentioning that the paper's technical contribution is minor since the main result is a direct consequence of van Trees' inequality. While this is not a problem in itself, it does not contribute to make the paper stand out.

**Questions:**

One of the key contributions of this paper is the introduction of dFIL as an intuitive privacy metric for instance encoding. However, it is important to note that, for (general) biased attacks, the information theorist's Fisher information plays an equally, if not more significant role than dFIL. Regrettably, the paper lacks a thorough discussion of the empirical and theoretical properties of the latter quantity, apart from its estimation using score matching. Consequently, it becomes challenging to evaluate the relative importance of dFIL compared to the information theorist's Fisher information. Clarifying these aspects would enhance the overall assessment of dFIL's significance.

Addendum: In their rebuttal, the authors clarified the relative importance of dFIL and the information theorist's Fisher information.

**Limitations:**

The authors addressed the limitations of the proposed method very well.

---

> ### Author Rebuttal · Authors · 2023-08-08
>
> We thank for the insightful review and feedback. We provide our answers to the questions and comments below. For the new evaluation data and their brief explanation, please check the global response.
>
> **Relative importance of dFIL compared to the information theorist's Fisher information**. While it is true that the information theorist’s Fisher information is important in deriving the bound, it is a quantity only related to the dataset, not the encoder design. It can be calculated once and shared with the community for each dataset, and also it can be refined over time within the community; each encoder designer does not have to calculate the information theorist’s Fisher information individually.
>
> Moreover, when comparing several encoder designs’ privacy under the same dataset, the information theorist’s Fisher information is just a constant, so calculating dFIL is more important in comparing each encoder’s relative privacy. Our work focuses on calculating dFIL, as our goal was to develop a metric that can be used to compare and evaluate the privacy of different encoders.
>
> **Minor technical contribution**. While our Corollary is directly based on the van Trees inequality, we are the first to connect the van Trees inequality to score matching in generative AI and use it to provide a bound for a real-world data distribution. Deriving any inferential privacy guarantee (bounding the posterior success rate of arbitrary adversaries) is known to be very challenging for real-world datasets with complex priors, and most existing works (on similar fields) assume simple data priors, such as data following Gaussian or uniform distribution [a, b]. We believe our insight in connecting the concept of score matching in generative AI is novel, and can be extended to other fields on inferential privacy.
>
> We also want to highlight that the theoretical result of our work, albeit simple, will have a high impact within the field. Currently, the field of instance encoding lacks **any** measure of privacy that is theoretically justified and empirically useful, despite the field’s high popularity. Our work is the **first** to bring theory to instance encoding privacy, which will inspire future researchers in the field to design encoders based on theoretical principles and deter bogus, empirical claims.
>
> [a] Arpita Ghosh et al., "Inferential privacy guarantees for differentially private mechanisms." Arxiv 2016.
>
> [b] Borja Balle et al., "Reconstructing training data with informed adversaries." S&P 2022.

---

> > ### Comment · Reviewer_AWfY · 2023-08-15
> >
> > Thanks a lot for your response, now it is easier to understand the relative importance of dFIL and the information theorist's Fisher information. I suggest you to add a comment in the paper on this matter. I updated my score.

---

### Official Review · Reviewer_yPPy · 2023-07-05

**Soundness:** 3 good
**Presentation:** 2 fair
**Contribution:** 2 fair
**Rating:** 5
**Confidence:** 2

**Summary:**

This paper discusses the privacy concerns surrounding data encoding methods employed in machine learning (ML) operations. Its primary objective is to present a theoretical framework that quantifies the extent of privacy leakage in an encoding scheme and facilitates the calculation of its invertibility. The authors propose a novel technique that leverages Fisher Information Leakage to compute the lower bound of Mean Squared Error (MSE) for an encoding scheme, considering both unbiased and biased attackers. Finally, The authors evaluate their technique against some simple reconstruction attacks both for training and inference tasks.


**Strengths:**

1. The paper takes a theoretical approach to explain reconstruction attacks. The problem is interesting and the presentation and flow of the paper are good. It was easy to follow and understand.
2. The authors' approach of initially focusing on unbiased attacks and subsequently extending their scheme to encompass attackers with prior knowledge is a good approach and beneficial in comprehending their contributions. The paper also provides evaluation of their technique considering both unbiased and biased attackers.
3. The paper is upfront about many of its limitations. This is commendable and helps understand the solution better and paves the way for identifying future research avenues.


**Weaknesses:**

1. The core contribution of the paper is adopting Fisher Information Leakage and van Trees inequality to come up with a lower bound for MSE. While this is a new idea, it would benefit from more comprehensive construction. Furthermore, the reconstruction attacks examined by the authors to showcase the efficacy of their technique are relatively simplistic. It would be advantageous for the authors to consider incorporating relevant literature from the privacy and security domain and evaluate their approach against more sophisticated and state-of-the-art reconstruction attacks.

2. The paper should provide more rigorous analysis of the privacy and utility trade-off. The paper readily discards Differential Privacy on utility ground, but later adopts noise addition to introduce privacy. Regardless, the privacy guarantee by noise addition and the corresponding utility trade-offs should be analytically explained. While the paper includes an empirical demonstration of the utility trade-off, it falls short of its intended goal of providing a theoretical explanation.

3. The authors should clarify how this lower bound contributes to preventing attacks, developing privacy-preserving encoding techniques, or addressing other important aspects related to data privacy.


**Questions:**

See weakness.

---

> ### Author Rebuttal · Authors · 2023-08-08
>
> We thank for the insightful review and feedback. We provide our answers to the questions and comments below. For the new evaluation data and their brief explanation, please check the global response.
>
> **Reconstruction attacks are simplistic**. We want to first highlight that the attacks we studied are based on *state-of-the-art works from top-tier conferences*. We tried both well-cited classics (e.g., CVPR ‘15 [34], CVPR ‘18 [35], ACSAC ‘19 [33]) and recent works (e.g., CCS ‘21 [36], CVPR ‘22 [20]), and chose the best-performing attacks. Especially, our attack-b in Figure 3--4 is directly adopted from CVPR ‘22 [20], which worked the best among the attacks we studied. The field does not have many sophisticated attacks because, unlike model inversion or gradient inversion which is much harder and still requires a lot of research, inverting instance encoding is easy, and existing simple attacks already work well in the absence of a defense like ours. Also, prior instance encoding works did not have theoretical justifications (unlike our work) and were considered not strong in terms of privacy [2, 23], so researchers had less motivation to design sophisticated attacks against them.
>
> Nonetheless, we agree with the reviewer that the work can benefit from a more sophisticated attack. Thus, we additionally designed an attack based on a DDPM diffusion model pretrained by Google [73]. Please refer to Appendix: Section 7.6 for more details. Appendix: Figure. 7 shows the reconstructed images with different dFIL. Similar to Figure 3-4, it can be seen that 1/dFIL >= 10 is relatively safe: pixel-to-pixel reconstruction becomes hard for 1/dFIL >= 6.42, and even the high-level semantic information becomes hard to reconstruct when 1/dFIL >= 49.5. At the same time, the figure shows that sometimes the high-level information of the image can be reconstructed (e.g., the fact that there is a white car facing left), even when our dFIL bound disallows exact pixel-by-pixel reconstruction. The result gives us an insight into what kind of information can still leak, even when a pixel-by-pixel reconstruction is prohibited by Corollary 1.
>
> We additionally plot what is equivalent to Figure 3 (left) for the diffusion model attacker in **Supplementary PDF: Figure 1**. The figure shows that the bound again works well even for the diffusion model attacker. Here, we used inputs with randomized smoothing noise as in Figure 3 to calculate the bound, and because the pretrained model is not trained with the randomized smoothing noise, we trained the diffusion model for 5000 epochs with the noisy inputs, following the hyperparameters from [73].
>
> **Why discard DP, but later adopts noise addition?** We want to clarify that adding noise is a privacy mechanism that is more general than DP, and our approach, although it adds noise, is not differentially-private. Prior work [2] theoretically proved that indistinguishability, which differential privacy aims to provide, is *fundamentally incompatible* with utility in instance encoding. As the privacy notion of indistinguishability is impossible to achieve, our approach is designed to achieve an alternative (weaker) privacy, non-invertibility. While it is also possible to achieve non-invertibility with DP methods, as DP was originally designed for indistinguishability, the privacy-utility trade-off becomes much worse. We show the empirical comparison with DP in Appendix: Section 7.5.
>
> **Lack of more rigorous theoretical analysis of the privacy and utility trade-off**. The main contribution of this paper is to (1) theoretically show that dFIL is a useful privacy metric that can bound the invertibility of instance encoding, and (2) empirically show that an instance encoding with a low dFIL can still have reasonable utility. Providing a theoretical analysis of the privacy-utility trade-off was not part of the scope of this paper, and we leave it as a future work.
>
> We believe our theoretical contribution is still meaningful, as we are the **first** work to provide a meaningful privacy metric that can bound the invertibility of an instance encoding. The result will help future researchers to design instance encoders using theoretical principles, rather than relying on empirical claims. Also, we do so by connecting the van Trees inequality with score matching in generative AI, a novel connection first made by our work. It allows bounding the posterior success rate of arbitrary adversaries for real-world data (e.g., CIFAR10) with intractable priors, which is known to be challenging in general. Prior works on similar fields usually assume data with simple priors like Gaussian or uniform [a, b]. We believe our idea of connecting score matching to capture real-world data priors can be extended to similar fields.
>
> **How does the lower bound contribute to developing privacy-preserving encoding?** dFIL allows us to quantitatively measure the invertibility of an instance encoding for the first time. System/model designers can now measure and compare their encoding techniques’ invertibility and select encodings that are harder to invert. This is exactly what we did in Section 4--5, where we tried out several empirically-proposed techniques from prior works  [17, 20, 22, 44, 58, 59] and chose techniques that provided the lowest dFIL (better privacy) with the same accuracy (Line 262--264). Before our work, there were no known ways to theoretically measure how private or not private each technique is, other than fully empirical methods.
>
> **The idea would benefit from more comprehensive construction**. We could not understand what this sentence meant exactly. Can you please elaborate, so that we can respond and/or add more results if necessary?
>
> [a] Arpita Ghosh et al., "Inferential privacy guarantees for differentially private mechanisms." Arxiv 2016.
>
> [b] Borja Balle et al., "Reconstructing training data with informed adversaries." S&P 2022.

---

> > ### Comment · Reviewer_yPPy · 2023-08-12
> >
> > Thanks for the detailed responses.
> > The concerns that I have about this paper are well addressed.
> > I will raise my score.

---

### Official Review · Reviewer_k1TB · 2023-07-05

**Soundness:** 3 good
**Presentation:** 3 good
**Contribution:** 3 good
**Rating:** 6
**Confidence:** 3

**Summary:**

Instance encoding (and some closely related lines of research) aim at finding ways to encode data examples (in a training set) as $E=Enc(e)$, in such a way that one can train models on the encoded examples $E_1,...E_n$, while $E_i$ does not leak much about $e_i$ to an adversary who inspects it.

Previous efforts for proposing "private" instance encodings have failed due to (1) subsequent attacks, and (2) lack of formal definitions of what privacy means here. In fact, it is known that such mechanisms cannot be DP in a strong sense, while they are also useful for training.

This paper aims to put forward a new way of arguing about privacy of instance encoding, based on Fisher Information (FI). FI is a useful measure of information revealed in random variables about a related random variable. In particular, knowing FI allows one to lower bound the Mean Squared Error (MSE) of the best reconstruction attack, assuming that the adversary has no prior knowledge about the distribution of the original instance.

More specifically, using FI, the paper shows how to give an *hardness of inversion* type privacy. The paper shows that both in the (more elementary) case of unbiased attacks (who see the original instances uniformly distributed) as well as more powerful attackers who have some prior information (e.g., know the distribution of the instances), one can lower bound the MSE of the best possible (information theoretic) attack, based on two parameters: FI of the encoding and a second parameter that depends on how flat/concentrated the distribution of the instance x is. Note that the second parameter also inherently shows up as one can approximate x well if it is too concentrated already.

The paper also discusses how to compute the two parameters of interest, or at least approximate them, in practical settings.
In particular, when the encoder is a Neural Net, the paper discusses how to use tools from previous work to approximate them using smooth functions that will be suitable for computing their FI.

The final product is an "average case" notion of privacy that is tailored for inputs coming from (flat enough) distributions.

The paper also presents experiments to validate its theory.

**Strengths:**

Differential privacy is the golden standard for privacy at the moment. But it is also a good idea to look for new notions privacy that could be meaningful on their own and useful for certain settings. Hence, I find the overall goal of the paper positive and useful for putting new perspectives on privacy on the table.

I also liked the fact that the paper made an effort to algorithmically compute their notion of privacy by approximating the two parameters of interest using tools from previous work.

**Weaknesses:**

(also a limitation) Bounding MSE sometimes might not say anything about privacy, when all an adversary wants to do is finding a specific sensitive feature (and not approximating the instance in $\ell_2$ norm). So, more (context dependent) work is needed for finding whether this notion is actually a good notion of privacy or not.

(also a question) The figures are not clear. For example, in Figure 2, what is the column numbers (Ref, 25, 10, 1) next to MNIST images?

(also a question and limitation) The paper does experiments in which the MSE of certain encoders are evaluated. However, it is not clear to me how *useful* these encodings are *for training models* over the encoded instances. Note that without knowing how useful the encodings are, there is always a trivial fully private encoding: output $\bot$ all the time.

**Questions:**

see the questions from the previous box.

is there any hope that your approach can say something about the expected $\ell_\infty$ distance between adversary's guess and the true instance (rather than MSE) ?

**Limitations:**

See the "weakness" section for some comments related to limitations as well.

On the positive side, the paper itself has a a good "limitations" section at the end.

---

> ### Author Rebuttal · Authors · 2023-08-08
>
> We thank for the insightful review and feedback. We provide our answers to the questions and comments below. For the new evaluation data and their brief explanation, please check the global response.
>
> **dFIL against sensitive attribute inference**. The initial motivation for dFIL was to protect against input reconstruction, which significantly hampers user privacy directly if allowed [28]. Still, dFIL can be adopted to protect sensitive attributes in some cases, if the attribute is contained within a subset of the input features. Currently, dFIL is an aggregate value across all features (e.g., pixels). Alternatively, we can calculate a per-feature FIL by looking at each entry in the diagonal of FIM instead of calculating the trace (Eq. 3) [27]. The per-feature FIL then gives us the reconstructability of each feature (pixel) individually. **Supplementary PDF: Figure 2** shows an example of this per-feature FIL for different images, for the split-middle architecture in Section 4. The figure shows that the key points of an image (e.g., contours of an object, tire of a car, tag of a cat, etc.) leak more through the encodings and can be easily reconstructed. If the sensitive attribute is contained in a subset of the image (e.g., eye color, background, …), we can measure its reconstructability directly using this per-feature FIL and understand if the attribute can be reconstructed by an adversary. If the sensitive feature is not contained within a few pixels (e.g., ethnicity), it would be harder to measure its leakage. A potential future direction is to disentangle the sensitive feature into a certain dimension in a latent space [a, b] and use dFIL in the latent space.
>
> **Figures are not clear (what are the column numbers in Fig 2?)**. The values indicate the 1/dFIL values (Line 231--232), which means more private if the value is higher. We will refine the figures in the final draft to make them more clear.
>
> **Utility of the encoders in Sec 3**. We perform two separate evaluations. First, in Section 3, we evaluate whether our MSE bound holds well against attacks. Second, in Section 4--5, we evaluate whether an encoder with a reasonably-low dFIL (high MSE bound) can still have good utility. We separate out the evaluations because we wanted to evaluate our bound in Section 3 against the most adversarial setup to see if it always holds.
>
> Encoders used in Section 3 were deliberately chosen to be simple and relatively easy to invert (e.g., single-layer convolution with a large output channel), and as they are designed solely for the ease of attack and not for utility, it is almost certain that they will not be very useful in training models. The goal of encoders in Section 3 was to show that the attackers cannot achieve an MSE below our bound *even for the encoders that are deliberately designed solely to make the attack easy (and do not care about utility)*. The results from Section 3 show that our MSE bound is reliable even in these adversarial cases.
>
> The second set of evaluations in Section 4--5 studies whether an encoder with a reasonably-low dFIL can still be useful in training/inference. For these encodings, we did not try inverting and comparing their MSE with the bound, as these practical encoders are already empirically harder to invert than the encodings that are deliberately made easy to invert (from Section 3), and we already showed that even the easier ones cannot be inverted to break the bound.
>
> **Can we bound L-inf?** While we cannot directly bound the expected L-inf, we can adopt the dFIL to bound something similar. Instead of calculating dFIL which is the average quantity of all features, we can calculate the expected squared L2 distance for each feature (the per-feature FIL discussed in the first bullet). If we take the max of these instead of an average, we can get the expected “squared” L-inf, which is similar (but not the same) with L-inf.
>
> [a] Ricky TQ Chen, et al. "Isolating sources of disentanglement in variational autoencoders." NeurIPS ‘18.
>
> [b] Zheng Ding, et al. "Guided variational autoencoder for disentanglement learning." CVPR ‘20.

---

> > ### Comment · Reviewer_k1TB · 2023-08-14
> > **thanks**
> >
> > thanks for the response.

---

### Official Review · Reviewer_KGoq · 2023-07-08

**Soundness:** 3 good
**Presentation:** 4 excellent
**Contribution:** 3 good
**Rating:** 7
**Confidence:** 3

**Summary:**

This paper introduces a new theoretical measure, the "diagonal Fisher Information Leakage" (dFIL), for quantifying the privacy leakage in instance encoding mechanisms in machine learning models. The authors construct a framework that balances the trade-off between data privacy and utility in instance encoding.

The authors start by developing the mathematical foundation of dFIL, leveraging the principle of Cramér-Rao Lower Bound (CRLB). They present a theoretical analysis which demonstrates that dFIL serves as an upper bound on the Mean Squared Error (MSE) of any estimator trying to reconstruct the original input from the encoded instances.

The paper presents two case studies to showcase the practical application of dFIL. The first one focuses on private inference, where instance encoding is applied to an existing pretrained model. The second case study explores the possibility of training a model on instance-encoded data. The results from these studies confirm that models guided by dFIL can achieve reasonable performance while ensuring a high degree of privacy.

Despite these advancements, the authors acknowledge the limitations of dFIL, such as the MSE bound not always correlating well with the semantic quality of the reconstruction, the average case bound provided by dFIL, and the difficulties of interpreting dFIL values in certain data types where MSE might not be directly meaningful or accurate. Recognizing these limitations, the paper calls for further research to explore and optimize the use of dFIL in designing and analyzing systems that use instance encoding for private inference and training.

**Strengths:**

Strengths:

1. **Originality**: The concept of Diagonal Fisher Information Leakage (dFIL) is novel, offering a fresh perspective on privacy leakage quantification in instance encoding for machine learning models. The authors have done a commendable job in providing a new metric that has potential broad applications in designing and analyzing private machine learning systems.

2. **Quality**: The theoretical foundation of dFIL is well-crafted and robust, using the Cramér-Rao Lower Bound to derive the maximum leakage. This gives dFIL a solid grounding in statistical theory. The authors carefully reason about the limitations of their approach and provide a roadmap for further exploration.

3. **Clarity**: The paper is well-written and structured, making the complex concept of dFIL comprehensible. The authors articulate the theoretical underpinnings and practical implications of dFIL with clarity and precision. The use of case studies adds to the understanding of how dFIL can be applied in practical scenarios.

4. **Significance**: The presented work is of significant value as it paves a way for protecting data privacy in instance encoding mechanisms, a critical issue in today's data-centric machine learning landscape. The demonstrated applications in private inference and training suggest the potential of dFIL to guide the design of machine learning systems that respect privacy constraints while maintaining utility.

**Weaknesses:**

1. **Limited empirical validation:** Although the theoretical development of dFIL is thorough and robust, the empirical validation seems somewhat limited. The authors conduct case studies on split inference and training with dFIL, which are commendable. However, additional experiments, especially on more diverse datasets and models, could strengthen the validity and generalizability of the dFIL measure.

2. **Absence of comparative analysis:** While the paper introduces dFIL as a novel concept, it doesn't provide a comparative analysis with other similar measures, if any exist. This makes it difficult for readers to evaluate the practical advantages or disadvantages of dFIL in contrast to other measures or approaches.

3. **Limited exploration of potential use-cases:** The authors present two case studies, however, the range of potential applications for dFIL could be broader. Additional use-cases illustrating how dFIL can guide the design of private machine learning systems would provide a more comprehensive view of the practical utility of dFIL.

4. **Interpretability of dFIL:** As the authors acknowledge in the Limitations section, for data types where MSE is not directly meaningful or the bound is inaccurate, interpreting the privacy of an encoding given its dFIL may not be straightforward. More investigation is needed to establish a clear and intuitive interpretation of dFIL values, especially in the context of real-world applications.

5. **Assumptions and constraints:** The assumptions underlying the derivation of dFIL and its applications need further exploration. For example, how sensitive is dFIL to the assumptions of model linearity, Gaussian noise, and others? The impact of violating these assumptions, and how they can be mitigated, would be valuable areas to explore.

**Questions:**

1. The paper uses mean squared error (MSE) as a metric to quantify the quality of reconstructions. However, as the authors acknowledge, this may not always correlate with the semantic quality of reconstructions. Could you please elaborate more on the choice of this metric and how it might affect the overall performance of the system? And the paper mentions that for certain data types where MSE is not directly meaningful, interpreting the privacy of an encoding given its dFIL might be challenging. Could you provide examples of such data types and explain how dFIL might be adapted to handle them?


**Limitations:**

The authors have done a commendable job addressing the potential limitations of their work. They detail four specific limitations in section 6:

1. dFIL only bounds the mean squared error (MSE), which may not always correspond well with the semantic quality of reconstructions.
2. The derived bounds are average-case, meaning that individual data instances could experience lower reconstruction MSE than the bound.
3. For data types where MSE is not directly meaningful or the bound is inaccurate, interpreting the privacy of an encoding given its dFIL might be difficult.
4. Systems with the same dFIL may have different invertibility, as the bound from dFIL could be conservative.

These limitations are articulated clearly, and the authors also suggest possible directions to address these limitations, such as exploring metrics other than MSE and calculating dFIL dynamically for each sample.

In terms of the broader societal impact, the paper does not directly address this aspect. However, it's understandable, given the theoretical nature of the work. Future research could consider potential misuse of this technology, as well as possible regulatory or ethical issues that might arise from its application, particularly in scenarios where private or sensitive data are involved. Nonetheless, the paper's focus on improving privacy in machine learning is itself an important societal contribution, as it addresses a growing need in an increasingly data-driven world.

---

> ### Author Rebuttal · Authors · 2023-08-08
>
> We thank for the insightful review and feedback. We provide our answers to the questions and comments below. For the new evaluation data and their brief explanation, please check the global response.
>
> **Comparison with similar measures**. Despite the popularity, there is *very little work on theoretical privacy analysis for instance encoding*, leaving us with no good alternative measures to compare. We discuss the lack of related works under the “Privacy metrics for instance encoding” paragraph in p. 2. The lack of theoretical measures in this popular field makes our work unique and important. We summarize the limitations of some of the related works below:
> - **Metrics without theory**. Some proposed *empirical* measures, such as mutual information (MI) [45] or distance correlation (dCorr) [21, 22]. However, these works failed to theoretically show how these measures are related to any form of privacy, if related at all (i.e., they lack what is often called an *operational interpretation*, which is considered essential in privacy literature [38]). As we explain in Line 95--98, prior works [38] warned against using such seemingly-valid but unjustified measures, because they can give a false sense of privacy. For example, [38] showed that MI sometimes mischaracterizes the severity of the information leakage (a system with a lower MI can actually be less private) and is not a good measure of privacy. As these measures (MI, dCorr) are not theoretically justified and can be misleading, we do not quantitatively compare them with dFIL. We will enrich Section 2 with the above qualitative discussion in the final draft.
> - **Metrics with theory**. There are metrics with operational interpretations; however, dFIL is much more practical than the others. Metric-DP [24] is not applicable to encoders used in Section 4 and 5 (Line 89). Differential privacy [30] degrades the utility too much (Line 82, Appendix: Section 7.5). A recent concurrent work proposed a new PAC privacy [a] that can bound the adversary’s success rate. However, the bound from [a] gets very loose on instance encoding setups we studied. For 1/dFIL=10 which was empirically hard to invert (Figure 5), the adversary attack success rate upper bound from [a] becomes 1 --- a trivial bound that does not give any practical information. We will add the discussion regarding [a] in the final draft.
>
> **Assumptions and constraints need further exploration**. dFIL is generally applicable to any differentiable and randomized methods, using the general Equation 1. The noise does not have to be Gaussian, and using different sources of randomization can lead to different privacy-accuracy tradeoffs. We leave such an exploration to future works. The differentiability (e.g., absence of ReLU or MaxPool) assumption is necessary for the van Trees inequality and the Cramer-Rao bound to hold. Although the bound may empirically work similarly without the differentiability assumption, we confined the scope of the paper to encoders where our Corollary can be mathematically precise. Please let us know of any other assumptions/constraints that the reviewer wants to discuss, if there are any.
>
> **Interpretability of dFIL**. We agree with the reviewer that interpreting dFIL might not be straightforward in some applications. As mentioned in Limitation 3 (p. 9), in such cases, acceptable values of dFIL should be determined for each application through further research. The well-adopted DP has similar issues: in many real-world scenarios, what $\epsilon$, $\delta$ means for a particular setting is not straightforward, and the acceptable values are often chosen empirically [69].
>
> **MSE as a metric**. For some data types (e.g., user’s geographical location), MSE can directly provide an intuitive notion of privacy. For other data types such as image or word embeddings, MSE might not exactly capture the semantic similarity of the reconstruction. In this work, we use MSE as a proxy to similarity as in prior works [47, 48], but the approach has limitations: two images with relatively high MSE may look semantically similar to humans. For example, see the images containing a white car on a red background in Appendix: Figure 7. Two images that are very different pixel-wise may still convey similar high-level information (e.g., the fact that there is a white car facing left in front of a red background), and MSE cannot capture the fact that such high-level information is still leaking, while it disallows an exact pixel-by-pixel reconstruction. The users of dFIL must understand these implications of dFIL and use it only when the privacy it provides makes sense. We believe our extensive discussions and evaluations will help future researchers to adopt dFIL in a meaningful way.
> As discussed in Limitation 1, van Trees inequality can be extended to potentially support metrics other than MSE (see Appendix: Theorem 2), which may be useful in some cases. We leave such an extension to future work.
>
> **Limited empirical validation**. We want to emphasize that our case study spans *three distinct applications (image classification, recommendation, and sentiment analysis)* that use *three different input data types (image, user behavioral features, and natural language)*, showing the generality of dFIL. Following the reviewers’ suggestion, we additionally add the CIFAR-100 dataset to the split inference evaluation in **Supplementary PDF: Table 1**. The trend is similar to Table 1 in the original paper.
>
> [a ] Hanshen Xiao, and Srinivas Devadas. "PAC Security: Automatic Privacy Measurement and Control of Data Processing." Crypto 2023 (to be appeared this August).

---

> > ### Comment · Reviewer_KGoq · 2023-08-10
> >
> > Dear Authors,
> >
> > Thank you for your detailed rebuttal in response to my review. I have received and read your explanations concerning the weaknesses, questions, and limitations raised. I will carefully consider your responses as I evaluate the revised version of the manuscript.

---

### Author Rebuttal · Authors · 2023-08-08

We thank the reviewers for their insightful reviews and feedback. We respond to the concerns and questions individually in a separate rebuttal for each review. Here, we upload a supplementary PDF containing additional results that were asked by the reviewers. We also give a brief explanation:

**Table 1** holds an additional evaluation result for split inference, using the CIFAR-100 dataset. The result directly extends Table 1 in the paper, and all the model hyperparameters are the same except for the last fully-connected layer. The result shows that our optimizations again improve the accuracy significantly on a reasonably-low dFIL (e.g., 1.98% -> 59.51%).

**Figure 1** holds a 1/dFIL vs. reconstruction MSE plot similar to Figure 3 (left) in the paper, for a new diffusion model-based attacker that we designed. The details of the diffusion model attacker are in Appendix: Section 7.6. To calculate the bound, we need to add a randomized smoothing noise to the input, so we retrained the model for 5000 epochs with the noise with the hyperparameters from Google [73]. The figure shows that the bound holds reliably for the diffusion model attacker as well.

**Figure 2** plots the per-feature FIL that can bound the reconstruction MSE of individual features (pixels), instead of the average of the entire input. Per-feature FIL can be calculated by directly using the diagonal entries of the Fisher information matrix (FIM), instead of calculating the trace (Eq. 3) [27]. Plotting the per-feature FIL helps us understand which pixels, or attributes occupying that pixels, can be more easily reconstructed. Figure 2 shows that identifying features, such as the contour of a frog (8th column), the tire of a car (7th column), or the necklace of a cat (9th column) leak more and can be reconstructed more easily.

---

### Decision · Program_Chairs · 2023-09-21

**Decision:**

Accept (poster)

**Comment:**

The paper proposes diagonal Fisher Information Leakage as a way to quantify the privacy leakage in instance encoding mechanisms. While  this notion has limitations, the paper does a reasonable job of discussing those. The reviewers found the work to be interesting and any concerns that were raised were largely addressed by the rebuttal. I would encourage the authors to update the paper to incorporate the reviewer feedback and the discussion. I am happy to recommend acceptance.